# The optical conductivity of few-layer black phosphorus by infrared spectroscopy

Guowei Zhang[1,2], Shenyang Huang[1], Fanjie Wang[1], Qiaoxia Xing[1], Chaoyu Song[1], Chong Wang[1], Yuchen Lei[1], Mingyuan Huang [3✉] & Hugen Yan [1✉]

The strength of light-matter interaction is of central importance in photonics and optoelectronics. For many widely studied two-dimensional semiconductors, such as $MoS_2$, the optical absorption due to exciton resonances increases with thickness. However, here we will show, few-layer black phosphorus exhibits an opposite trend. We determine the optical conductivity of few-layer black phosphorus with thickness down to bilayer by infrared spectroscopy. On the contrary to our expectations, the frequency-integrated exciton absorption is found to be enhanced in thinner samples. Moreover, the continuum absorption near the band edge is almost a constant, independent of the thickness. We will show such scenario is related to the quanta of the universal optical conductivity of graphene ($\sigma_0 = e^2/4\hbar$), with a prefactor originating from the band anisotropy.

[1] Department of Physics, State Key Laboratory of Surface Physics and Key Laboratory of Micro and Nano Photonic Structures (Ministry of Education), Fudan University, 200433 Shanghai, China. [2] MIIT Key Laboratory of Flexible Electronics (KLoFE) and Xi'an Institute of Flexible Electronics, Northwestern Polytechnical University, 710072 Xi'an, Shaanxi, China. [3] Department of Physics, Southern University of Science and Technology, 518055 Shenzhen, China. ✉email: huangmy@sustech.edu.cn; hgyan@fudan.edu.cn

In recent years, two-dimensional (2D) materials including graphene, transition metal dichalcogenides (TMDCs) and black phosphorus (BP), are at the forefront of scientific research. Strong light-matter interactions have been demonstrated in these atomically thin materials[1–12], holding great promise for photonic and optoelectronic applications. Optical absorption in 2D materials is a fundamental light-matter interaction process, basically governed by the optical sheet conductivity $\sigma(\hbar\omega)$. Graphene is a well-known 2D example, which exhibits a universal conductivity $\sigma_0 = e^2/4\hbar$ in a broad frequency range, where $e$ is the electron charge and $\hbar$ is the reduced Plank constant[1,2]. Moreover, $N$-layer graphene exhibits an optical conductivity of $N\sigma_0$, showing quantized optical transparency[1,13]. In semiconducting TMDCs, such as $MoS_2$, the optical conductivity due to K-point exciton increases with layer number[14]. However, in this paper, we will show that excitons in thinner samples absorb more light in few-layer BP. Meanwhile, the absorption due to the electron-hole continuum near its own edge is almost the same for each thickness, with a value close to that in monolayer graphene.

Few-layer BP is an elemental 2D semiconductor beyond graphene, with a strongly layer-dependent direct bandgap[15–18], offering an ideal platform to probe the layer-dependent properties and dimensional crossover from 3D to 2D. Moreover, the intrinsic in-plane band anisotropy definitely distinguishes few-layer BP from other widely studied 2D materials, such as graphene, TMDCs and InSe. Along with the moderate bandgap and large tunability, few-layer BP is unique and promising in polarized IR detectors and emitters. From this point of view, a quantitative determination and thorough understanding of the optical absorption in few-layer BP is in great demand. The optical absorption of few-layer BP is found to be dominated by robust excitons[10–12]. In previous optical studies[10,15,16], the absorption intensity is not well quantified. In other words, quantitative insight of the absolute absorption is yet to be gained, though it is highly desirable for future optoelectronic applications, such as evaluating the quantum efficiency of photoluminescence (PL) and photocurrent generation. Our study resolves this issue and provides new insights into light-matter interaction in anisotropic 2D gapped materials.

## Results

**Sample preparation and IR characterization**. A Fourier transform infrared (FTIR) spectrometer was used to obtain the extinction spectra $(1 - T/T_0)$ of few-layer BP on polydimethylsiloxane (PDMS) or quartz substrates (see Methods for sample preparation and IR characterization), where $T$ and $T_0$ denote the light transmittance of the substrate with and without BP samples, respectively, as illustrated in Fig. 1d. For atomically thin materials on a thick transparent substrate, like in our case, when the optical conductivity is not large, the extinction $(1 - T/T_0)$ is approximately proportional to (the real part of) the optical conductivity[2,4]: $\sigma(\hbar\omega) = (1 - T/T_0) \cdot (n_s + 1) \cdot c/8\pi$, where $n_s$ is the refractive index of the substrate ($n_s = 1.39$ for PDMS in the measured IR range[19]) and $c$ is the speed of light.

We systematically measured IR conductivity $\sigma(\hbar\omega)$ spectra (in the unit of $\sigma_0$) for 2–7 L BP in a broad range of 0.4–1.36 eV at room temperature, as shown in Fig. 2. The incident light is normal to the layer plane and polarized along the armchair (AC) direction. The spectra for zigzag (ZZ) polarization are featureless[15], hence not discussed here. Previous studies have revealed quantized subband structures of few-layer BP[15,16], due to the quantum confinement in the out-of-plane direction and considerable interlayer interactions, in analogy to traditional quantum wells (QWs). In symmetric QWs with normal light incidence, optical transitions obey the $\Delta j = 0$ selection rule ($j$ is the subband index)[20,21]. $E_{jj}$ denotes the exciton resonance associated with the optical transition between the $j$th pair of subbands ($v_j \rightarrow c_j$) at $\Gamma$ point of the 2D Brillouin zone, as illustrated in Fig. 1b. As seen from Fig. 2, the $E_{11}$ resonance exhibits a very narrow linewidth even at room temperature, especially for thicker BP. In addition, the Stokes shift of few-layer BP is almost negligible (see Supplementary Fig. 1 as an example of 3 L), indicating good sample quality. The IR conductivity $\sigma(\hbar\omega)$ at the $E_{11}$ resonance reaches $6.6\sigma_0$ in 2 L BP, directly translated to a light absorption of 15% in free-standing case. This suggests very strong light-matter interactions in this atomically thin material. Moreover, we can observe a spectrally flat and broad response above the exciton energy, attributed to the continuum of band-to-band transitions. Step-like features of continuum absorption underline the step-like 2D joint density of states (DOS) in QW-like structures, as sketched in Fig. 1c, this still holds in anisotropic few-layer BP[22].

**Layer-dependent exciton absorption**. The $\sigma(\hbar\omega)$ spectra can provide a wealth of information on the exciton oscillator strength, which is directly related to the frequency-integrated conductivity (or absorption) of excitons[23,24], as indicated by the shaded areas in Fig. 2. The peak height of the exciton feature is not as informative, since it is sensitive to the sample quality. Figure 3a shows the integrated conductivity $\sigma_I$ of the ground state (1 s) exciton of few-layer BP as a function of layer number $N$. The details for spectral fitting and extraction of $\sigma_I$ are presented in Methods. For each layer thickness, at least three samples were measured, with the error bar defined as the spread of the data. From Fig. 3a, one can find that thinner BP has larger absorption. In other words, it manifests a fact that less material absorbs more light at exciton resonances. This remarkable result is in sharp contrast to widely studied 2D semiconducting TMDCs, in which the exciton absorption increases with layer number (see Supplementary Fig. 2 for the absorption of 1–4 L $MoS_2$ and also ref. [14]). To be more quantitative, the integrated absorption due to excitons is directly proportional to $L_z \cdot |\varphi_{ex}(0)|^2$, where $L_z$ is the thickness of the sample, $|\varphi_{ex}(0)|^2$ is the modular square of the exciton wavefunction at origin, describing the probability to find the electron and the hole at the same location[23–25]. Certainly, the smaller the thickness is, the larger the confinement in the $z$ direction is and hence the larger $|\varphi_{ex}(0)|^2$ is. Theory has predicted that $|\varphi_{ex}(0)|^2 \sim 1/L_z^2$ for QWs[26], if the penetration of the electron and hole wavefunctions into the barriers is negligible[27,28]. This gives the integrated absorption $\sim 1/L_z$ ($L_z \sim N$ for BP), which means stronger absorption for thinner samples. This argument correctly predicts the trend but the fitting with such a relation does not work well. The prediction shows a much steeper decrease than what we observed, as shown in Fig. 3a (black dashed curve), which calls for a more refined model.

We can tackle this problem from another perspective. In the 2D hydrogen model for excitons, we find that the integrated absorption for the 1 s exciton is proportional to the exciton binding energy ($E_b$) (see Supplementary Note 1 for details). This is a very reasonable conclusion, since the larger the binding energy is, the closer the electron is to the hole, which favors light absorption at exciton resonances. We adopt this result for our analysis, even though excitons are not ideally 2D in few-layer BP. Olsen et al. proposed a simple screened hydrogen model for 2D excitons[29], in which $E_b$ can be analytically expressed as a function of the reduced effective mass $\mu$ and the 2D sheet polarizability $\chi$. Under the condition of $32\pi\mu\chi/3 >> 1$, $E_b$ can be simplified as $E_b \approx 3/4\pi\chi$, which is proved to be valid for few-layer BP[10]. The sheet polarizability $\chi$ relates to the dielectric screening of electron-hole

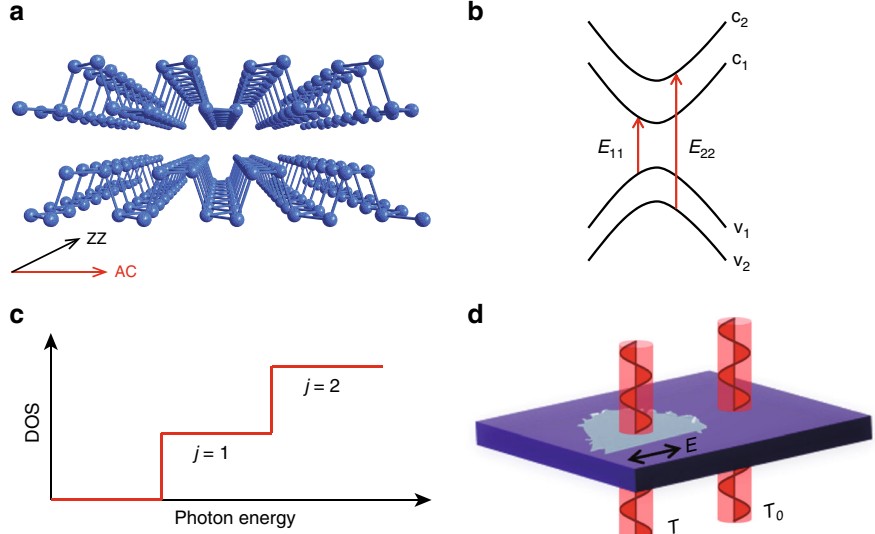

**Fig. 1 Crystal structure and IR characterization of few-layer BP. a** Puckered structure of 2 L BP, with two characteristic crystal orientations: AC and ZZ. **b** Schematic illustration of optical transitions between quantized subbands, with the symbols $E_{11}$ and $E_{22}$ denoting transitions $v_1 \rightarrow c_1$ and $v_2 \rightarrow c_2$, respectively. **c** Illustration of joint DOS in 2D systems, showing step-like features. **d** FTIR measurement configuration with $T$ and $T_0$ denoting the light transmittance of the substrate with and without BP samples, respectively. The arrow indicates the polarization of the incident IR beam, along the AC direction of the BP sample.

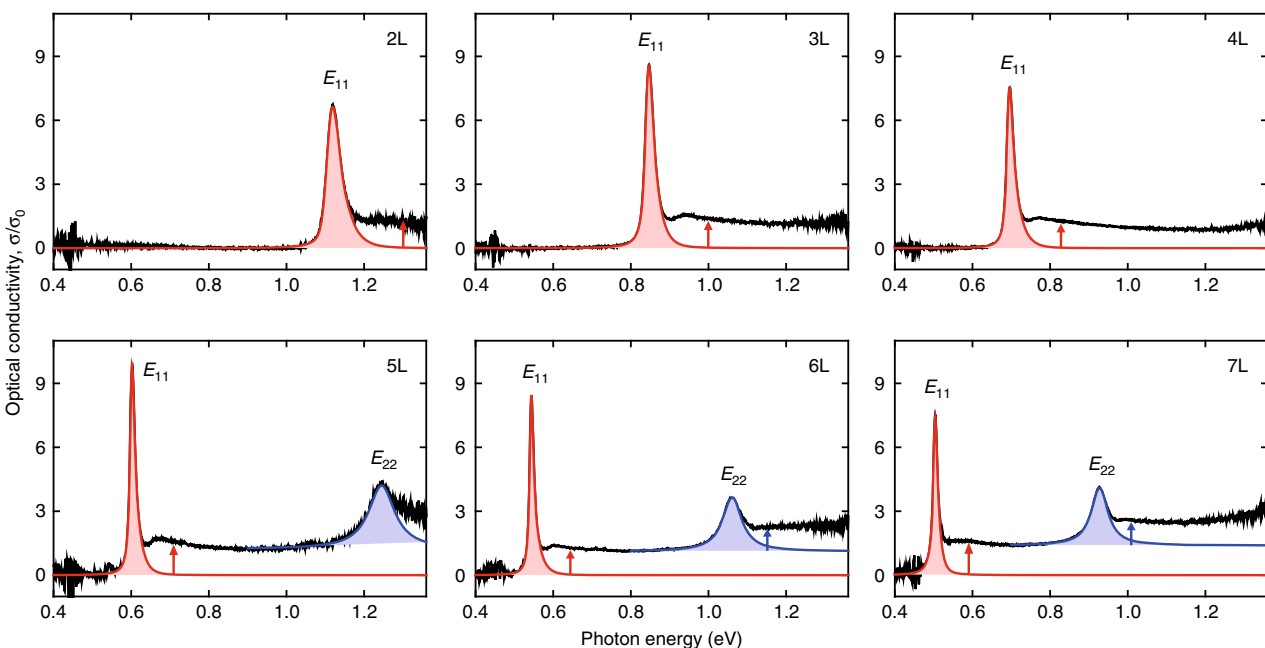

**Fig. 2 Room-temperature IR measurements.** Optical conductivity $\sigma(\hbar\omega)$ spectra of few-layer BP with layer number $N = 2$–$7$, with incident light polarized along the AC direction, in the unit of $\sigma_0 = e^2/4\hbar$. $E_{11}$ ($E_{22}$) denotes the $1s$ excitonic transition associated with the first (second) pair of subbands, as illustrated in Fig. 1b. The slightly asymmetric $E_{11}$ peaks are fitted using Eq. (3) in Methods (red curves), while $E_{22}$ peaks are fitted to symmetric Lorentzian lineshapes (blue curves). The exciton absorption (or conductivity $\sigma_I$) is proportional to the integrated areas of the exciton peaks, as indicated by the shaded areas. The vertical arrows indicate the height of the continuum absorption for each subband transition.

interactions. In the case of Fig. 2, few-layer BP is supported by a PDMS substrate, which introduces additional dielectric screening. Thus, both of the substrate and the material itself contribute to the screening. In view of this, the sheet polarizability $\chi$ is replaced by an effective value for $N$-layer BP, expressed as $\chi_{\mathrm{eff}} = \chi_0 + N\chi_1$, with $\chi_0 = 6.5\,\text{Å}$ and $\chi_1 = 4.5\,\text{Å}$ describing the screening effect from the PDMS substrate and single-layer BP, respectively, based on our previous study[10]. Thus, we have $\sigma_I \propto 1/(\chi_0 + N\chi_1)$. With this relation, it is clear that the exciton absorption increases as the

layer number decreases, though it is not as dramatic as $\sim 1/N$. We use this relation to fit the data for $E_{11}$, as shown in Fig. 3a, the overall agreement is much better than the $1/N$ scaling. The basic behavior is well captured by the modified model, though the agreement with experimental data is still not so excellent. The deviation is mainly caused by the experimental uncertainty in determining the optical conductivity, especially for thinner BP samples, which are expected to be more susceptible to the environment. The substrate plays a role in reducing the exciton

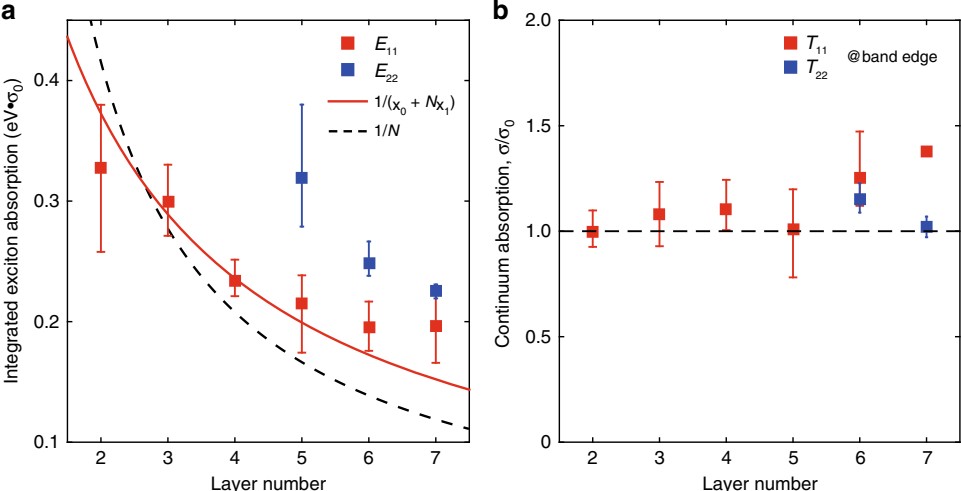

**Fig. 3 Exciton and continuum absorption in few-layer BP. a** The frequency-integrated conductivity $\sigma_I$ of the $1s$ exciton as a function of layer number, in the unit of eV·$\sigma_0$. For each layer number, at least three samples were measured, the squares denote the average values. The solid curve is the fitting to the $E_{11}$ data using the relation $\sigma_I = h/(\chi_0 + N\chi_1)$, with $\chi_0$ and $\chi_1$ describing the screening effect from the PDMS substrate and single-layer BP, respectively. $h$ is the only fitting parameter, extracted to be 5.78 eV·$\sigma_0$·Å. The dashed curve shows the $1/N$ relation for guideline. **b** The continuum height of each subband transition near the corresponding band edge ($\hbar\omega \approx E_g$) is plotted as a function of layer number. The continuum absorption nearly shows a constant value, independent of the material thickness. The black dashed line indicates the well-known universal conductivity $\sigma_0 = e^2/4\hbar$ for guideline. Data are extracted from Fig. 2 with error bars indicating sample-to-sample variation.

binding energy, hence the integrated absorption. For freestanding BP, this model also gives a scaling of $1/N$ (ref. [10]), consistent with the previous prediction for QWs. For $E_{22}$ resonances, the exciton absorption increases with the decrease of layer number as well, as also shown in Fig. 3a. It should be noted that, the thickness dependence of exciton absorption in BP is even qualitatively quite different from that in traditional QWs. For the latter, the absorption reaches maximum at certain thickness and decreases again with decreasing thickness, due to the exciton wavefunction leakage into the barriers in the ultrathin limit[27,28]. Such scenario becomes more evident in shallow QWs. The hard confinement in atomically thin BP definitely distinguishes it from traditional QWs. From this perspective, atomically thin BP provides us unique opportunities to examine this new type of QWs.

The frequency-integrated conductivity of excitons is robust against temperature, although the linewidth is typically vulnerable to various factors. Since the aforementioned PDMS substrate thermally expands (contracts) during the heating (cooling) process, significant strain effects can be introduced to few-layer BP samples during temperature-dependent measurements, overwhelming the pure temperature effect. Therefore, we transferred few-layer BP samples to quartz substrates with a much smaller thermal expansion coefficient, so that the strain effect can be ignored[30]. Fig 4a, d show the IR extinction spectra of a 3 L and 7 L BP at varying temperatures from 10 K to 300 K, respectively. To improve the ratio of signal-to-noise for such measurements, the incident IR beam size was set to be larger than the sample size. Thus, the absolute intensity of exciton absorption is underestimated, but it's legitimate to compare the integrated area of exciton peaks for the same sample at different temperatures, which is directly related to the exciton oscillator strength. To quantitatively probe the temperature effect, the integrated area and linewidth of exciton peaks are extracted from spectral fitting using Eq. (3) in Methods. Clearly, the exciton width increases with temperature both for $E_{11}$ and $E_{22}$ as expected (Fig. 4c, f). Such scenario is common in semiconductors, which is attributed to the enhanced electron-phonon scattering channels at elevated temperature[31,32]. The integrated areas of exciton peaks, on the

other hand, are nearly independent on the temperature, as shown in Fig. 4b, e. The slight deviation from a constant may arise from the experimental uncertainty and data fitting procedure. A recent study shows that for typical semiconductors, in the incoherent region of light-matter interactions, the integrated exciton absorption only depends on the radiative decays, rather than the scattering decays, hence the absorption is independent of temperature[33]. This is exactly the case for our observations. Since the integrated exciton intensity is proportional to the exciton binding energy, our results indicate that the exciton binding energy is almost a constant within the tested temperature range.

**Continuum absorption.** Next, we will focus on the layer dependence of the absorption due to the continuum transition. As seen in Fig. 2, a relatively flat response can be observed in the $\sigma(\hbar\omega)$ spectra above the ground state exciton energy, mainly associated with the continuum band-to-band transitions (labeled as $T_{jj}$) and possibly excited excitonic states[10]. A closer examination of Fig. 2 shows that the continuum absorption is not strictly a constant but decreases gradually with increasing photon energy. For simplicity, we focus on the continuum absorption close to the band edge ($\hbar\omega \approx E_g$), indicated by the red (blue) arrows for $T_{11}$ ($T_{22}$) transitions in Fig. 2. The results are summarized in Fig. 3b as a function of layer number $N$, they approximately follow a constant value despite of the experimental uncertainty. This means that the absorption due to the bandgap continuum transitions is almost the same, regardless of the thickness of BP samples.

According to the Fermi's golden rule, if a light beam with frequency $\omega$ and linear polarization along $\vec{e}_0$ is normally incident to a direct-gap 2D system, the dimensionless absorption $A(\hbar\omega)$ due to a pair of conduction and valence bands can be expressed as (in SI units)[34]

$$A(\hbar\omega) = \frac{\pi e^2}{n_s c \varepsilon_0 m_0^2 \omega} \frac{1}{A_r} \sum_{\vec{k}} \left| \vec{P}_{cv}(\vec{k}) \right|^2 \delta(E_{cv} - \hbar\omega) \qquad (1)$$

where $\varepsilon_0$ is the vacuum permittivity, $m_0$ is the free electron mass, and $A_r$ is the area of the 2D material. The momentum matrix

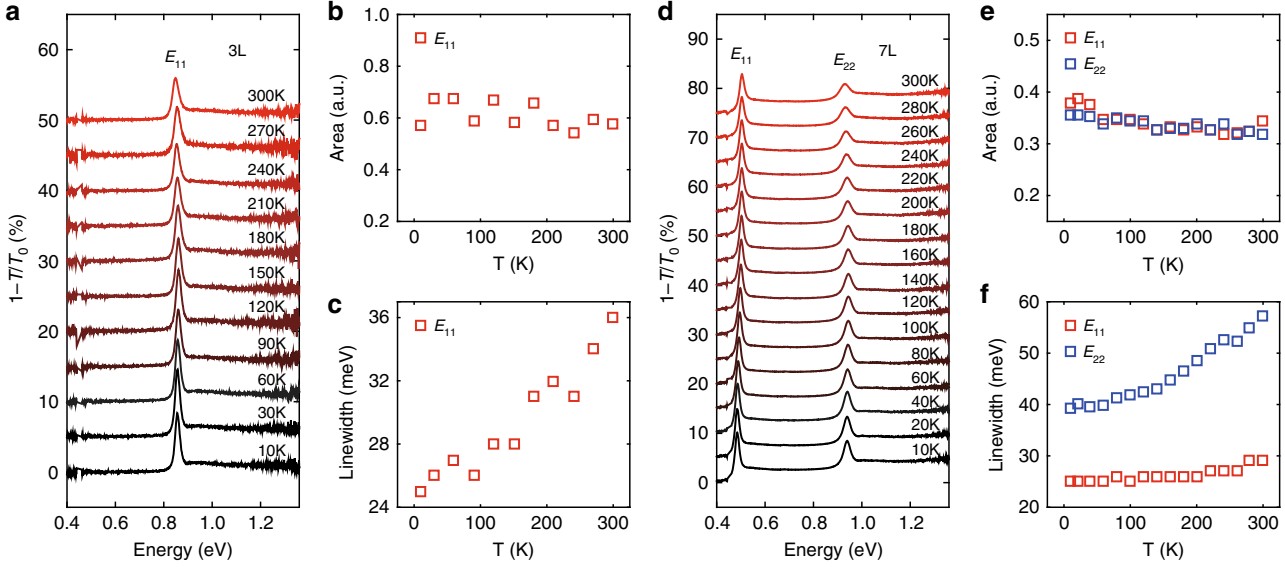

**Fig. 4 Temperature dependence of the optical absorption.** IR extinction spectra of **a** 3 L and **d** 7 L BP on quartz substrates at varying temperatures from 10 K to 300 K, respectively. The spectra are vertically offset for clarity. Integrated area of the exciton peaks of **b** 3 L and **e** 7 L BP as a function of temperature, respectively. Linewidth of the exciton peaks of **c** 3 L and **f** 7 L BP as a function of temperature, respectively. Data are extracted from spectral fitting using Eq. (3) in Methods.

element is defined as $\overrightarrow{P}_{cv}(\overrightarrow{k}) = \langle c, \overrightarrow{k} | \overrightarrow{e}_0 \cdot \overrightarrow{P} | v, \overrightarrow{k} \rangle$, with $\overrightarrow{P} = -i\hbar \nabla$ as the momentum operator. $E_{cv}(\overrightarrow{k}) = E_{c\overrightarrow{k}} - E_{v\overrightarrow{k}}$ and $\overrightarrow{k}$ is the wave vector.

It first looks like that the absorption is strongly band parameter dependent, since the terms of $\overrightarrow{P}_{cv}(\overrightarrow{k})$ and DOS are involved in Eq. (1). Surprisingly, the absorption in isotropic graphene[1,2] and InAs QWs[35] are found to be universal, indicating that there is a cancelling mechanism between these two terms. For massless graphene, which has a unique linear energy dispersion near the Dirac point $E_{cv}(k) = \pm \hbar v_F k$ ($v_F$ is the Fermi velocity), the cancelation is exact with no frequency dependence, leading to a well-defined universal absorption quanta $\pi\alpha$, where $\alpha = e^2/4\pi\varepsilon_0 \hbar c$ is the fine structure constant (~1/137)[36]. For anisotropic 2D massive semiconductors (see Supplementary Note 2 for details), based on the $\overrightarrow{k} \cdot \overrightarrow{p}$ perturbation theory[34], it is obtained that

$$E_{cv}(\overrightarrow{k}) = E_g + \frac{\hbar^2}{2}\left(\frac{k_x^2}{\mu_x} + \frac{k_y^2}{\mu_y}\right), \text{ and } \frac{1}{\mu_x} = \frac{4|\overrightarrow{P}_{cv}|^2}{m_0^2 E_g}, \text{ where } \mu_{x(y)} \text{ is the}$$

reduced effective mass in the AC (ZZ) direction. With these, and also taking into account the degrees of spin and valley degeneracy of $g_s$ and $g_v$, we thus have

$$A(\hbar\omega) = \frac{g_s g_v \pi\alpha}{2} \cdot \frac{E_g}{\hbar\omega} \cdot \sqrt{\frac{\mu_y}{\mu_x}} \cdot \Theta(\hbar\omega - E_g) \quad (2)$$

where $\Theta(\hbar\omega - E_g)$ is the step function, describing the 2D joint DOS, and $E_g$ is the bandgap. This indicates that most of the band parameter dependent terms are perfectly canceled out, leaving behind the frequency dependent term $\omega$ and the band anisotropy term $\sqrt{\frac{\mu_y}{\mu_x}}$. Our observations in few-layer BP can be well explained by Eq. (2): (i) above the bandgap $\hbar\omega > E_g$, the absorption gradually decreases as the photon energy increases; (ii) near the bandgap $\hbar\omega \approx E_g$, with $g_s = 2$, $g_v = 1$ for BP[20], Eq. (2) gives a value of $\pi\alpha\sqrt{\frac{\mu_y}{\mu_x}}$ (or equals to the optical conductivity $\sigma_0\sqrt{\frac{\mu_y}{\mu_x}}$), slightly deviating from the universal value $\pi\alpha$. The prefactor $\sqrt{\frac{\mu_y}{\mu_x}}$ originates from the band anisotropy, which is nearly layer-independent for 2–7 L BP[17]. When $\mu_x = \mu_y$, it collapses to the isotropic case[36]; (iii) with the photon energy higher than the second subband transition ($T_{22}$), additional contribution in absorption shows up with another step height of $\pi\alpha\sqrt{\frac{\mu_y}{\mu_x}}$.

## Discussion

As a simple explanation of what we observed for both exciton and electron-hole continuum transitions, we can revisit the 2D band structure of few-layer BP. Due to the strong coupling between layers, the conduction and valence bands split into multiple 2D subbands with sizable energy spacing (for the case with sample thickness below 10 L)[15]. If we focus on the bandgap continuum transition, no matter the thickness or how many subbands in total, the relevant ones are only $c_1$ and $v_1$ subbands. As a consequence, one can not expect more absorption with photon energy right above the bandgap when the layer number increases, given that only the first pair of 2D subbands is involved and the 2D joint DOS typically barely vary. Therefore the increase of sample thickness does not cause enhanced absorption for the continuum. However, on the other hand, the excitonic effect weakens with increasing layer number due to weaker confinement in the z direction, which actually decreases the exciton absorption. This can qualitatively explain the findings in Fig. 3. This argument applies to other 2D materials as well. For MoS₂, the electronic bands associated with direct-gap exciton at K point show almost no splitting when the layer number increases from 1 to 2 (or other thickness)[37]. The degenerate subbands certainly double the DOS and the exciton absorption can increase accordingly, giving an opposite trend compared to BP. As for N-layer graphene, in spite of the complexity of the very low energy band structure, in the range from visible to near-IR, N pairs of subbands are involved in the optical absorption and each pair contributes an absorption quanta $\pi\alpha$[1,2]. While for N-layer BP, if we only focus on the bandgap region, only one pair of subbands is involved and one absorption step is contributed, regardless of the thickness. Of course, if we increase the photon energy, more and more subbands are counted in and the absorption increases step by step. Eventually the continuum absorption is comparable to that of N-layer graphene. Therefore, counting the 2D subbands

does give us a good estimation of absorption in different 2D systems[36].

In summary, we have determined the optical conductivity of 2–7 L BP using IR absorption spectroscopy. Our results reveal that the exciton absorption increases as the layer number decreases, a direct consequence of enhanced excitonic effects in reduced dimensionality. Moreover, the absorption from the continuum states near the band edge exhibits a layer-independent value. Few-layer BP provides us an ideal platform to probe the dimensional effect on the strength of optical absorption from bound (exciton) and unbound (continuum) states in the same material. The highly enhanced exciton absorption of atomically thin BP, which tends to host high density of optical excitations, may open up new possibilities for applications in nonlinear optics and quantum optics. Our results are expected to stimulate further theoretical interests in anisotropic 2D materials.

## Methods

**Sample preparation**. Few-layer BP samples were prepared by a modified mechanical exfoliation method[38]. Firstly, a piece of bulk BP crystal (HQ Graphene Inc.) was cleaved several times using a Scotch tape. Secondly, the tape containing BP flakes was slightly pressed against a PDMS substrate of low viscosity and then peeled off rapidly. Some thin BP flakes with relatively large area and clean surface were left on the PDMS substrate. To achieve high optical quality, the samples on PDMS were directly used for room-temperature IR measurements, without any additional sample transfer process. Supplementary Fig. 1 shows an example of a 3 L BP on PDMS. The negligible Stokes shift indicates good sample quality. To avoid sample degradation in air, the samples were prepared in a $N_2$ glove box with $O_2$ and $H_2O$ levels lower than 1 ppm, and then loaded into a cryostat purged with $N_2$ for subsequent IR measurements. The cryostat is only for sample protection instead of temperature control. The layer number and crystal orientation of BP were readily identified using polarized IR spectroscopy[15].

**Polarized IR spectroscopy**. The IR extinction $(1 - T/T_0)$ spectra of few-layer BP were measured using a Bruker FTIR spectrometer (Vertex 70 v) equipped with a Hyperion 2000 microscope, as illustrated in Fig. 1d. A tungsten halogen lamp was used as the light source to cover the broad spectral range of 0.4–1.36 eV with 1 meV resolution, in combination with a liquid nitrogen cooled Mercury-Cadmium-Telluride (MCT) detector. The lower bound cutoff photon energy is restricted by the substrate. The linearly polarized incident light was obtained after the IR beam passing through a broadband ZnSe grid polarizer, then was focused on BP samples using a ×15 IR objective. For room-temperature measurements, the aperture size was set to be smaller than the sample size to make sure the IR beam all goes through the sample. A spectrum was typically acquired over 1000 averages to improve the signal-to-noise ratio.

**Low temperature IR measurements**. For low temperature measurements, few-layer BP samples were transferred from PDMS to quartz substrate, to avoid the significant strain effect of PDMS during the heating (cooling) process, since the latter has a much smaller thermal expansion coefficient. Then, the samples were loaded into a liquid He cooled cryostat, with the temperature range from 10 K (or lower) to 300 K. To improve the signal-to-noise ratio, the aperture size was set to be larger than the sample size.

**Data analysis**. Generally, homogeneous broadened lineshape is characterized by a Lorentzian function. However, for the $E_{11}$ peaks in Fig. 2, attempts failed to fit the lineshape using a single Lorentzian function, since the high-energy tail deviates the spectrum from the ideal lineshape, especially for thinner BP. Similar observations have been reported in QWs[39–41], the exciton absorption (or emission) lineshape is slightly asymmetric, exhibiting a Lorentzian and an exponential slope at the low- and high-energy side, respectively. This is caused by the so-called "exciton locali-zation effect", as a consequence of spatial inhomogeneity[39]. Nevertheless, such imperfection will not affect our determination of the integrated areas of exciton peaks.

To determine the integrated conductivity $(\sigma_I)$, the $E_{11}$ peaks in Fig. 2 are fitted using a documented model, which was first proposed by Schnabel et al.[39], and has been successfully applied to QW-like structures[40,41]. This model is analytically expressed as:

$$\sigma(\hbar\omega) = \frac{L_S}{\pi} \frac{1}{\gamma_L + \Delta^2/\gamma_L} + \frac{L_A}{2\eta} \left[ 1 + \mathrm{erf}\left(\frac{\Delta}{\gamma} - \frac{\gamma}{2\eta}\right) \right] \cdot \exp\left(\frac{\gamma^2}{4\eta^2} - \frac{\Delta}{\eta}\right) \quad (3)$$

The first part is a Lorentzian function, describing the symmetric lineshape with area of $L_S$ and linewidth of $\gamma_L$. $\Delta = \hbar\omega - \hbar\omega_0$, with $\hbar\omega_0$ denoting the average transition energy between the conduction and valence subbands. The second part describes the asymmetric lineshape, $\gamma$ is the linewidth and $\eta$ describes the

asymmetric broadening by exciton localization, $L_A$ is the spectral weight. We use Eq. (3) to fit the $E_{11}$ peaks. As shown in Fig. 2 (red curves), the general agreement is excellent, except for a small discrepancy at the high-energy end of the peaks, given that the excited excitonic states (2 s or 3 s states) and the continuum absorption are also involved in this spectral region. While $E_{22}$ peaks (blue curves) can be well fitted only using the symmetric part (Lorentzian function). The integrated conductivities shown in Fig. 3a are extracted from the fitted peaks. In addition, as indicated by the extracted parameter $\eta$ for each layer thickness (Supplementary Fig. 4), we can see that thinner BP exhibits larger asymmetry. This is reasonable, since thinner BP is more sensitive to the underlying substrate and environment.

## Data availability

The data that support the findings of this study are available from the corresponding author upon reasonable request.

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

## Acknowledgements
H.Y. is grateful to the financial support from the National Young 1000 Talents Plan, National Science Foundation of China (grants: 11874009, 11734007), the National Key Research and Development Program of China (grants: 2016YFA0203900 and 2017YFA0303504), Strategic Priority Research Program of Chinese Academy of Sciences (XDB30000000), and the Oriental Scholar Program from Shanghai Municipal Education Commission. M.H. is grateful to the financial support from the Initiative Funds for Shenzhen High Class University (grant: G02206301). G.Z. acknowledges the financial support from the National Natural Science Foundation of China (grant: 11804398), Natural Science Basic Research Program of Shaanxi (grant: 2020JQ-105), Open Research Fund of State Key Laboratory of Surface Physics and the Fundamental Research Funds for the Central Universities. Part of the experimental work was carried out in Fudan Nanofabrication Lab.

## Author contributions
H.Y. and G.Z. initiated the project and conceived the experiments. G.Z. prepared the samples and performed the room-temperature IR measurements. G.Z., M.H., and S.H. performed the low temperature IR measurements with assistance from F.W., Q.X., C.S., C.W., and Y.L. G.Z., M.H., and H.Y. performed the data analysis and co-wrote the manuscript. H.Y. supervised the whole project. All authors commented on the manuscript.

## Competing interests
The authors declare no competing interests.
