## [Peer Review File · Nature Communications]

Reviewers' comments:

Reviewer #1 (Remarks to the Author):

The paper discussed the optical conductivity of black phosphorus vs a number of layers. Authors found that exciton absorption is enhanced for single-layer black phosphorus.

The data are thoroughly analyzed and the paper is properly prepared, so it is recommended to be accepted with minor corrections as stated below:

1. The different substrates for BP transfer were used to apply low-temperature studies. Nevertheless, they induce different stress in the sample due to vdW and adhesion. Stress at the specific direction will modify the DoS of samples inducing also different optical properties. Please comment on that. Please show the reference quartz data studied at RT by polarized IR spectroscopy.
2. Please expand the experimental part of how the sample was fabricated and positioned to position it normally to the armchair direction. This is missing in methods. Particularly, in the low-temperature IR studies experiment.
3. Authors have observed specific tail in the E11 peaks not fitting to Lorentzian. Could you elaborate more and support that with some estimation or simulation?
4. The conclusion that exciton leakage into the barriers for specific thickness should be supported by some longer discussion. Please refer to some DFT studies or conduct own DFT experiment to support your observation.
5. In general, DFT simulation studies will considerably support the stated conclusions and formed hypotheses in the discussion.

Overall, the paper is well organized and prepared, and I recommended to publish it after addressing the above issues.

Reviewer #2 (Remarks to the Author):

Using infrared spectroscopy, the authors study the optical conductivity of few-layer black phosphorus (BP), demonstrating that the frequency-integrated exciton absorption increases as the number of layers decreases, while the continuum absorption near the band edge is constant independent of the number of layers. Since 2D materials such as few-layer black phosphorus have recently attracted considerable attention, the presented optical study seems timely.

In my opinion, however, there are several points in the manuscript which are not clear enough. Here I list some of them which should be clarified or improved more in the revised version, as explained below.

- The main features of few-layer BP presented in this paper such as the exciton and universal continuum absorptions are not unique characteristics of few-layer BP. Rather these features are shared by other 2D materials such as quantum wells and graphene, as mentioned by the authors. Only exception could be TMDCs such as MoS₂, which shows the enhanced exciton absorption as the number of layers increases. In this sense, it is not clear why few-layer BP is interesting or has special optical properties compared with other 2D materials.

- Detailed comparisons with other experimental results such as Refs. 10, 15, 16 and with theoretical calculations such as Refs. 11, 12, 19, 20 are required, even though the authors

mentioned in the main text that “the absorption intensity is not well quantified”. Especially, Ref. 10 by Zhang et al seems to present very similar results with those presented in the current work, especially Figs. 2 and 3 in Ref. 10.

- In the manuscript, a simple 2D electron gas model with an isotropic effective mass is used for the theoretical analysis. However, Few-layer BP is highly anisotropic and cannot be simply approximated as a 2D electron gas over a broad range of energy. For example, in Fig. 1c, the constant jointed DOS is assumed for each interband transition and the universal continuum absorption is based on this constant effective mass approximation, which is valid only over the limited energy range. The effective Hamiltonian for few-layer BP describing the low-energy conduction and valence bands is well known, for example, as shown in Eq. (1) of Baik et al, Nano Lett. 15, 7788 (2015). The analysis based on the improved model could be more convincing.

- In line 121, the authors compare the absorption between traditional QWs and few-layer BP, arguing that the “hard confinement” in atomically thin BP distinguishes it from the QWs. What is the reason for the hard confinement in few-layer BP? Is it unique in few-layer BP?

- In Eq. (2) of the main text and Supplementary Note 2, the authors present the result of the universal absorption based on Supplementary Ref. 6. Is it valid for any massive 2D systems irrespective of their Dirac nature? It would be useful to discuss the validity of the result.

- If possible, it could be interesting to apply a perpendicular electric field which could tune the energy gap even to a negative value inducing a phase transition, as demonstrated in the above mentioned Baik et al, Nano Lett. 15, 7788 (2015) and Fig. 2 there. Or even in the insulator phase with a finite electric field, there appear additional interband transitions which are not allowed in the absence of an electric field, as demonstrated by Lin et al of Ref. 20. It could be interesting to study how the additional peaks and corresponding excitation/continuum absorptions evolve with the number of layers in the presence of a perpendicular electric field.

Reviewer #3 (Remarks to the Author):

The manuscript reported an increasing exciton absorption and a constant continuum absorption in few-layer BP as the layer number decreased. The latter finding has been extensively reported for various 2D structures, including 2D layered materials and traditional quantum wells. The former finding is remarkably different from other 2D layered materials due to quantum confinement and discrete subbands, which is of great interest to researchers in low-dimensional physics. The following concerns need to be addressed before consideration for publication.

1. In Fig. 2, the width of the exciton peaks increased as the layer number decreased, which seems to contribute to the most increasing of the integrated absorption. Could the authors explain this?

2. In Fig. 3a, the authors added the substrate screening to decrease the slope. But the fitting is still not good enough for the more slowly descending data points.

3. For the vertical arrows, how did the authors choose the position of the arrows? The reviewer suggests to fit the continuum absorption and get the optical conductivity at the band edge, which would make Fig. 3b more convinced.

4. For the title. “Less is more” is only correct for the increasing exciton absorption, not for the constant continuum absorption. The present title may produce misleading.

Response to the reviewers' comments

Reviewer #1 (Remarks to the Author):

The paper discussed the optical conductivity of black phosphorus vs a number of layers. Authors found that exciton absorption is enhanced for single-layer black phosphorus. The data are thoroughly analyzed and the paper is properly prepared, so it is recommended to be accepted with minor corrections as stated below:

Reply: We thank the reviewer for his/her careful review of our manuscript, particularly appreciate the reviewer's recommendation of publication in Nature Communications after minor corrections.

1. The different substrates for BP transfer were used to apply low-temperature studies. Nevertheless, they induce different stress in the sample due to vdW and adhesion. Stress at the specific direction will modify the DoS of samples inducing also different optical properties. Please comment on that. Please show the reference quartz data studied at RT by polarized IR spectroscopy.

Reply: We agree with the reviewer that the substrate will modify the optical properties or optical conductivity, for the following reasons: 1) As the reviewer said, the vdW interactions and adhesion between samples and different substrates are different, so that the stress induced by the substrate is different. It has been demonstrated that stress (or strain) has a pronounced effect on the optical properties of few-layer BP (Nat. Commun. 2019, 10, 2447). As shown in Fig. R1, tensile strain induces significant blueshift of E_{11} exciton peak, but the strength (i.e., area of the exciton peak) is barely affected within the tested strain range. 2) For 2D materials, due to additional dielectric screening of electron-hole interactions, the substrate plays a key role in reducing the exciton binding energy (Sci. Adv. 2018, 4, eaap9977), and hence the strength of exciton absorption. In the measured IR spectral range (0.4-1.36 eV), the refractive index of PDMS and quartz substrates is approximately the same ($n_{\text{PDMS}} \approx 1.39$ and $n_{\text{quartz}} \approx 1.42$). Thus, we propose that the two

substrates have little difference regarding on the exciton absorption strength. 3) Most of all, the BP transfer from PDMS to other substrates (e.g., quartz, BaF₂) will cause damage in sample quality, partially (or mainly) for the reason that after the sample transfer to new substrates, some traces of adhesive from PDMS will leave on the surface of samples. As shown in Fig. R2, the E_{11} exciton absorption is obviously reduced for samples on quartz or BaF₂ substrates compared with those on PDMS, since the exciton absorption is very sensitive to sample quality. According to our experience, samples on PDMS are of the highest quality, thus samples exfoliated on PDMS are directly used for IR measurements at room temperature without any transfer process.

Besides, PDMS is not suitable for low-temperature experiments, since strain will be induced during the heating (cooling) process. Thus, samples are transferred to quartz substrate for low-temperature studies. Nevertheless, we'd like to emphasize that it's still fair to compare the exciton absorption at different temperatures for the same sample.

Fig. R1: (a) IR extinction ($1-T/T_0$) spectra of a 6L BP on polypropylene (PP) substrate under different biaxial strains, the spectra are vertically shifted for clarity; (b) The E_{11} peaks are horizontally shifted, indicating that the area of exciton peaks are the same under different strains. (adapted from Nat. Commun. 2019, 10, 2447).

Fig. R2: Optical conductivity of 6L and 4L BP on different substrates at room temperature.

2. Please expand the experimental part of how the sample was fabricated and positioned to position it normally to the armchair direction. This is missing in methods. Particularly, in the low-temperature IR studies experiment.

Reply: The samples were fabricated using a modified mechanical exfoliation method (2D Mater. 2014, 1, 025001) in a glove box filled with N₂. Firstly, a piece of bulk BP crystal (commercially obtained from HQ Graphene Inc.) was cleaved several times using a Scotch tape. Secondly, the tape containing BP flakes was slightly pressed against a PDMS substrate of low viscosity and then peeled off rapidly. In this way, some thin BP flakes with relatively large area and clean surface were left on the PDMS substrate, which can be readily distinguished from thick ones by optical contrast using an optical microscope.

The crystal orientation of few-layer BP was determined by FTIR-based polarized IR spectroscopy. The original polarization of the broadband IR beam from a tungsten halogen lamp is in all directions. A linear polarization is obtained when the IR beam passes through a ZnSe grid polarizer. According to our previous study (Nat. Commun. 2017, 8, 14071), the IR absorption of BP is strongly polarization dependent: the strongest absorption corresponds to the AC polarization and the nearly zero absorption for the ZZ polarization (see Fig. R3). Just by rotating the polarizer, we can easily position BP normally to the AC direction.

In the low-temperature IR experiments, BP samples were fabricated and positioned using the same method described above. The only difference is that the samples were transferred to quartz substrates by a dry transfer method, and then loaded into a cryostat. This is still done in a glove box.

In the revised manuscript, we made more detailed description in the Methods part as follows: (added sentences/words are marked in red, deleted ones are marked by the strikethrough)

Page 11, Lines 237-241: *"Few-layer BP samples were prepared by a modified mechanical exfoliation method³⁸. Firstly, a piece of bulk BP crystal (HQ Graphene Inc.) was cleaved several times using a Scotch tape. Secondly, the tape containing BP flakes was slightly pressed against a PDMS substrate of low viscosity and then peeled off rapidly. Some thin BP flakes with relatively large area and clean surface were left*

on the PDMS substrate. ~~mechanically exfoliated on transparent PDMS substrates from bulk crystals (HQ Graphene Inc.).~~

Page 12, Lines 254-256: *“The linearly polarized incident light was obtained after the IR beam passing through a broadband ZnSe grid polarizer, then was focused on BP samples using a 15X IR objective. The incident light was focused on BP samples using a 15X IR objective, with its polarization controlled by a broadband ZnSe grid polarizer.”*

Fig. R3: Polarization dependent IR extinction spectra of a 6L BP, 0° (90°) corresponds to the AC (ZZ) direction (adapted from Nat. Commun. 2017, 8, 14071).

3. Authors have observed specific tail in the E11 peaks not fitting to Lorentzian. Could you elaborate more and support that with some estimation or simulation?

Reply: As shown in Fig. 2 of the main text, we can see that the E_{11} peaks are asymmetric, with a long tail at the high-energy side, which deviates the spectra from the ideal Lorentzian lineshape, especially for thinner BP. Similar observations have been reported in quantum wells (Ref. 39-41), caused by the so-called “exciton localization effect”, as a consequence of the spatial inhomogeneity (Ref. 39). In our case, the few-layer BP samples are large, typically over 20 μm in size. In such large spatial range, the “inhomogeneity” is inevitable, very likely due to the inhomogeneous strain from the underlying substrate.

The fitting parameter η in Eq. (3) of the main text describes the asymmetric broadening by exciton localization, which is summarized in Fig. R4 for each layer number. It indicates that the thinner the sample is, the more asymmetric the lineshape is. This is quite reasonable, since thinner samples are more sensitive to the influence from the underlying substrate and environment.

In the revised manuscript, we made corrections in the Methods part as follows: (added sentences/words are marked in red, deleted ones are marked by the strikethrough)

Page 13, Lines 286-288: *“In addition, as indicated by the extracted parameter η for each layer thickness (Supplementary Fig. 4), we can see that thinner BP exhibits larger asymmetry. This is reasonable, since thinner BP is more sensitive to the underlying substrate and environment.”*

Fig. R4: The fitting parameter η as a function of layer number. η is extracted from the fitting of E_{11} peaks using Eq. (3) of the main text, describing the asymmetric broadening by exciton localization. Error bars are from sample-to-sample variations.

4. The conclusion that exciton leakage into the barriers for specific thickness should be supported by some longer discussion. Please refer to some DFT studies or conduct own DFT experiment to support your observation.

Reply: The hard confinement in atomically thin BP definitely distinguishes it from traditional QWs. For the latter, the exciton wavefunction leaks into the barriers in the ultrathin QW limit, leading to the non-monotonic scaling of exciton absorption with thickness. Here, we'll present detailed discussions.

After extensive studies of transitional QWs by several groups in the past experimentally and theoretically, now it's clear how the exciton oscillator strength evolves within a broad thickness range (from tens of nanometers to a couple of nanometers). One typical experimental result is shown here (Fig. R5(a)). We can see

that the oscillator strength first increases, and then decreases as the well thickness decreases. The oscillator strength reaches maximum at thickness around 5-7 nm (depending on the type of the QW material). The non-monotonic scaling behavior in transitional QWs has also been verified by theoretical calculations, as shown in Fig. R5(b), in the thickness range of 1-30 nm (indicated by the two red lines), the trend qualitatively agrees with the experimental results. This is in sharp contrast to few-layer BP, in which the exciton absorption strength always increases with the decreasing thickness (Fig. 3(a) of our manuscript), even in the one or two layer limit. While for traditional QWs with thickness in the range of 2-7L BP (1-3.5 nm), the oscillator strength decreases with decreasing thickness.

The different behaviors originate from different quantum confinement. For few-layer BP, the confinement can be regarded as infinitely high QWs and the wavefunction of the carriers can not leak out. However, for traditional QWs, the barrier is not high (finitely high) and the wavefunction can leak into the barrier region, especially when the well is very thin. Therefore, from the intermediate thickness (a few tens of nm) to the ultrathin (a couple of nm) regimes, the exciton strength shows opposite trends. In the intermediate thickness regime, the thickness of the well is around the exciton Bohr radius (a_B) and the leakage of the carrier wavefunction can be neglected. In this regime, a decreased thickness will enhance the electron and hole overlap, hence increases the oscillator strength. While in the ultrathin regime, the Bohr radius is larger than the thickness and the leakage of the wavefunction into the barrier regime becomes more and more severe with decreasing thickness, resulting in a decreasing electron-hole overlap and hence a reduced oscillator strength.

In the revised manuscript, we cited a theory paper to support the discussion regarding the exciton leakage in traditional QWs, as follows:

Page 6, Lines 130-131: “..., due to the exciton wavefunction leakage into the barriers in the ultrathin ~~QW~~ limit^{27, 28}.”

[28] Iotti, R. C. & Andreani, L. C. Crossover from strong to weak confinement for excitons in shallow or narrow quantum wells. *Phys. Rev. B* **56**, 3922 (1997).

Fig. R5: (a) Experiment: measured well thickness dependence of the exciton oscillator strength in $\text{In}_x\text{Ga}_{1-x}\text{As}/\text{GaAs}$ QWs (adapted from PRB 1994, 50, 7499); (b) Theory: calculated well thickness dependence of the exciton oscillator strength in $\text{GaAs}/\text{Al}_{0.15}\text{Ga}_{0.85}\text{As}$ QWs (adapted from PRB 1997, 56, 3922).

5. In general, DFT simulation studies will considerably support the stated conclusions and formed hypotheses in the discussion.

Reply: From our point of view, if experimental results can be explained reasonably well using simple theoretical models, we prefer not resorting to DFT calculations. DFT does give lots of detailed results. However, it tends to lose insight in many cases. Therefore, in our current study, we decide not to present DFT results. Instead, we use well-accepted exciton models and $k \cdot p$ method to capture our major findings and cite related theory papers to support our conclusions. Our work is expected to stimulate more theoretical interests in the future.

Overall, the paper is well organized and prepared, and I recommended to publish it after addressing the above issues.

Reply: We thank the reviewer again for his/her positive assessment of our work and recommendation of publication in Nature Communications.

~~~~~

**Reviewer #2 (Remarks to the Author):**

Using infrared spectroscopy, the authors study the optical conductivity of few-layer black phosphorus (BP), demonstrating that the frequency-integrated exciton absorption increases as the number of layers decreases, while the continuum absorption near the band edge is constant independent of the number of layers. Since 2D materials such as few-layer black phosphorus have recently attracted considerable attention, the presented optical study seems timely.

In my opinion, however, there are several points in the manuscript which are not clear enough. Here I list some of them which should be clarified or improved more in the revised version, as explained below.

*Reply: We thank the reviewer for his/her careful review of our manuscript. And we made modifications in the revised manuscript according to the reviewer's comments.*

- The main features of few-layer BP presented in this paper such as the exciton and universal continuum absorptions are not unique characteristics of few-layer BP. Rather these features are shared by other 2D materials such as quantum wells and graphene, as mentioned by the authors. Only exception could be TMDCs such as MoS2, which shows the enhanced exciton absorption as the number of layers increases. In this sense, it is not clear why few-layer BP is interesting or has special optical properties compared with other 2D materials.

*Reply: This is a good question. We will explain in more detail here why few-layer BP is unique for this study, compared with other 2D structures, such as transitional quantum wells (QWs), and the most widely studied 2D materials – graphene and TMDCs (MoS2 et al.)*

*1) Comparison with transitional QWs. We'd like to point out that the thickness dependence of exciton strength in few-layer BP is fundamentally different from that in traditional QWs. In fact, in the ultrathin limit, the trend is just opposite. After extensive studies of QWs by several groups in the past (PRB 1994, 50, 7499; PRB 1997, 56, 3922), now it's clear how the exciton oscillator strength evolves within a broad thickness range (from tens of nanometers to a couple of nanometers). One*

typical result is shown here (Fig. R6(a)). We can see that the oscillator strength first increases, and then decreases as the well thickness decreases. The oscillator strength reaches maximum at thickness around 5-7 nm (depending on the type of the QW material). This is in sharp contrast to few-layer BP, in which the exciton absorption strength always increases with the decreasing thickness (Fig. 3(a) of our manuscript), even in the one or two layer limit. While for traditional QWs with thickness equivalent to 2-7L BP (1-3.5 nm), the oscillator strength decreases with decreasing thickness.

The different behaviors originate from different quantum confinement. For few-layer BP, the confinement can be regarded as infinitely high QWs and the wavefunction of the carriers can not leak out. However, for traditional QWs, the barrier is not high (finitely high) and the wavefunction can leak into the barrier region, especially when the well is very thin. Therefore, from the intermediate thickness (a few tens of nm) to the ultrathin (a couple of nm) regimes, the exciton strength shows opposite trends. In the intermediate thickness regime, the thickness of the well is around the exciton Bohr radius ( $a_B$ ) and the leakage of the carrier wavefunction can be neglected. In this regime, a decreased thickness will enhance the electron and hole overlap, hence increases the oscillator strength. While in the ultrathin regime, the Bohr radius is larger than the thickness and the leakage of the wavefunction into the barrier regime becomes more and more severe with decreasing thickness, resulting in a decreasing electron-hole overlap and hence a reduced oscillator strength.

Some of the previous studies (PRB 1985, 32, 8027; PRB 1985, 32, 4275), only observed an increase of the exciton strength with decreasing thickness, one example is shown in Fig. R6(b). That's because they are still in the intermediate thickness regime. In that regime, the exciton strength does qualitatively behave similarly to that of few-layer BP. However, quantitatively, those two cases are still different, with the exciton oscillator strength per unit area for the former  $\sim 1/L_z$  ( $L_z$  is the well thickness) and for the latter  $\sim 1/(L_z + \chi)$ , with  $\chi$  accounting for the substrate screening effect, as demonstrated in this study. Most importantly, if those traditional QWs have a thickness of few-layer BP (several nm), the behavior is opposite to that of few-layer BP since they already enter the ultrathin regime.

2) *Comparison with graphene and TMDCs.* Graphene is semimetallic, excitons are not hosted in it due to strong dielectric screening of electron-hole interactions. Hence, graphene is repulsive to steady state exciton physics. For TMDCs (such as

MoS2), as the reviewer said, the optical absorption due to K-point exciton increases with layer number, showing an opposite trend compared with BP. As seen in Supplementary Note 3, we can understand this from the band structure. In bilayer MoS2, the conduction and valence bands at K-point do not split even with layer-layer interactions. As a consequence, the joint density of states is doubly degenerate at K point in bilayer. Similarly, it is N-fold degenerate in N-layer. Therefore thicker MoS2 absorbs more light.

In brief summary, we now see that the exciton behavior of few-layer BP is truly different from that of excitons in traditional QWs, which shows non-monotonic scenario. The layer dependence of exciton strength in few-layer BP is also distinctively different from that in TMDCs. Our study presents a nice example of excitons in exfoliated 2D semiconductors and largely complements the previous knowledge of 2D excitons. As a matter of fact, it's not easy to find another system with hard confinement (infinite well) which can so cleanly manifest the exciton strength evolution with thickness, particularly, down to the atomic layer thickness.

Fig. R6: (a) Well thickness dependence of the exciton oscillator strength in InxGa1-xAs/GaAs QWs with the thickness in the ultrathin and intermediate regime (adapted from PRB 1994, 50, 7499); (b) Well thickness dependence of the exciton oscillator strength in GaAs/AlAs QWs with the thickness only in the intermediate regime (adapted from PRB 1985, 32, 4275).

- Detailed comparisons with other experimental results such as Refs. 10, 15, 16 and with theoretical calculations such as Refs. 11, 12, 19, 20 are required, even though the authors mentioned in the main text that “the absorption intensity is not well

quantified". Especially, Ref. 10 by Zhang et al seems to present very similar results with those presented in the current work, especially Figs. 2 and 3 in Ref. 10.

Reply: It's a good advice to compare the present results with previous experimental and theoretical results in terms of optical studies of few-layer BP.

1) *Experimental studies*. Detailed comparisons are shown in Fig. R7 and Table 1. Definitely, the samples in this work are of the ever highest quality. Besides, as the reviewer said, the absorption spectra in our previous study (Ref. 10, Sci. Adv. 2018, 4, eaap9977) and this work look very similar to each other at first sight, we show them in the same figure (Fig. R8) for further quantitative comparison. We'd like to clarify here that we can't extract the optical conductivity from previous data, thus we can't quantitatively compare the absolute absorption intensity of different layer number. For example, the previous 4L, 5L and 6L data (red curves) exhibit a lower exciton absorption than those for the present ones (black curves). The reason is that in previous IR measurements, the aperture size is deliberately set to be larger than the sample size to improve the signal to noise ratio. As a consequence, the absolute absorption is underestimated. That's why we say in the introduction part of our manuscript that "*In previous optical studies10, 15, 16, the absorption intensity is not well quantified.*". To extract the absolute absorption (or optical conductivity), and then to compare them with different layer thickness, in this work, the aperture size is set to be smaller than the sample size to ensure that the IR beam can all go through the sample. To ensure the data repeatability, at least three samples are measured for each layer number (2-7L).

2) *Theoretical studies*. Rodin et al. (Ref. 11, PRB 2014, 90, 075429) and Chaves et al. (Ref. 12, PRB 2015, 91, 155311) focus on the exciton side. Rodin et al. calculated the exciton binding energy of 1L BP, especially highlighted the relation between exciton binding energy and substrate dielectric constant, this agrees well with our assumption that substrate adds additional screening and thus reduces the exciton effects. Chaves et al. studied the excitonic properties of few-layer BP under in-plane electric field. We extract the zero-field values, shown in Fig. R9. It clearly shows that the oscillator strength in 1-4L BP increases with decreasing thickness, and exhibits a linear relation with exciton binding energy. This agrees well with our findings. Low et al. (Ref. 19, PRB 2014, 90, 075434) and Lin et al. (Nano Lett. 2016, 16, 1683) focus on the continuum side, they studied thick BP films, in which excitonic effects are excluded. Fig. R10 shows the calculated optical conductivity of 4-20 nm BP by Low et

al., in general, it agrees nicely with our findings: a) the first continuum step exhibits a value of  $\sim\sigma_0$  for 4, 6, 8, 10 nm BP, the second step contributes to another  $\sim\sigma_0$ , approximately independent of the thickness; b) above the bandgap, the conductivity decreases with frequency. Lin et al. calculated the optical conductivity of BP films (>4 nm, or 8 layers) in the presence of a perpendicular electric field. They found that under sufficient electric field strength, forbidden transitions ( $v_2$  to  $c_1$ , or  $v_1$  to  $c_2$ ) will occur, with part of oscillator strength transferred from the main transitions ( $v_1$  to  $c_1$ ,  $v_2$  to  $c_2$ ).

Fig. R7: Comparison of previous studies and this work, taking a 4L BP as an example.

Fig. R8: Present (black, this work) and previous (red, Sci. Adv. 2018, 4, eaap9977) IR data of 2-7L BP, shown in the same figure for quantitative comparison.

Table 1: Comparison of present results with previous experimental results of few-layer BP

|                      | Ref. 16 (Nat. Nanotechnol. 2017, 12, 21) | Ref. 15 (Nat. Commun. 2017, 8, 14071) | Ref. 10 (Sci. Adv. 2018, 4, eaap9977) | This work                    |
|----------------------|------------------------------------------|---------------------------------------|---------------------------------------|------------------------------|
| Layer number         | 1-5                                      | 2-15                                  | 2-8                                   | 2-7                          |
| Spectral range       | Visible to near-IR (0.75-2.5 eV)         | Near to mid-IR (0.25-1.36 eV)         | Near to mid-IR (0.4-1.36 eV)          | Near to mid-IR (0.4-1.36 eV) |
| Substrate            | sapphire                                 | quartz                                | PDMS                                  | PDMS                         |
| Sample quality       | Low                                      | Low                                   | High                                  | Highest                      |
| Optical conductivity | ×                                        | ×                                     | ×                                     | ✓                            |

Fig. R9: Calculated oscillator strength of 1s excitons in few-layer BP as a function of layer number and exciton binding energy (adapted from PRB 2015, 91, 155311).

Fig. R10: Calculated optical conductivity of 4-20 nm BP (adapted from PRB 2014, 90, 075434).

- In the manuscript, a simple 2D electron gas model with an isotropic effective mass is used for the theoretical analysis. However, Few-layer BP is highly anisotropic and cannot be simply approximated as a 2D electron gas over a broad range of energy. For example, in Fig. 1c, the constant jointed DOS is assumed for each interband transition and the universal continuum absorption is based on this constant effective mass approximation, which is valid only over the limited energy range. The effective Hamiltonian for few-layer BP describing the low-energy conduction and valence bands is well known, for example, as shown in Eq. (1) of Baik et al, Nano Lett. 15, 7788 (2015). The analysis based on the improved model could be more convincing.

Reply: Thanks very much for the reviewer's suggestion. Firstly, We'll show that the step-like density of states (DOS) still holds in anisotropic 2D materials. Let's consider a 2D system with area of  $S = L_x \cdot L_y$ , the effective masses are denoted by  $m_x$  and  $m_y$  ( $m_x \neq m_y$ ), and wave vectors by  $k_x$  and  $k_y$  in the x and y directions, respectively. The energy dispersion for free electrons is expressed as:

$$E(k) = \frac{\hbar^2 k_x^2}{2m_x} + \frac{\hbar^2 k_y^2}{2m_y} \quad (R1)$$

In the polar coordinate system, Eq. (R1) can be rewritten as:

$$E(k) = \frac{\hbar^2 k^2}{2} \left( \frac{\cos^2 \theta}{m_x} + \frac{\sin^2 \theta}{m_y} \right) \quad (R2)$$

According to Eq. (R2), we have

$$dk = \frac{dE}{\hbar^2 k \left( \frac{\cos^2 \theta}{m_x} + \frac{\sin^2 \theta}{m_y} \right)} \quad (\text{R3})$$

The number of states between E and E+ΔE is

$$\Delta Z = D(E)dE = \frac{S}{2\pi^2} \int_0^{2\pi} k dk d\theta \quad (\text{R4})$$

Note that the spin degeneracy of 2 is already included in Eq. (R4).

Inserting (R3) into (R4), the expression of density of states D(E) is obtained,

$$D(E) = \frac{S}{2\pi^2 \hbar^2} \int_0^{2\pi} \frac{d\theta}{\frac{\cos^2 \theta}{m_x} + \frac{\sin^2 \theta}{m_y}} \quad (\text{R5})$$

The integral gives a result of  $2\pi\sqrt{m_x m_y}$ , then we have

$$D(E) = \frac{S\sqrt{m_x m_y}}{\pi \hbar^2} \quad (\text{R6})$$

With  $m_x = m_y = m$ , Eq. (R6) gives the result of isotropic 2D cases,  $D(E) = \frac{Sm}{\pi \hbar^2}$ .

Eq. (R6) clearly tells us that DOS in anisotropic 2D systems is a constant as well, independent of energy, resembling the step-like features in DOS of isotropic 2D systems.

Secondly, we thank the reviewer very much for reminding us of the band anisotropy nature of BP. In the revised manuscript, we made modifications in the theoretical analysis of continuum absorption (Supplementary Note 2), with the band anisotropy effect taken into consideration. Based on the improved model, we find that the band anisotropy indeed modifies the optical absorption, contributing to a

prefactor  $\sqrt{\frac{\mu_y}{\mu_x}}$  ( $\mu_x$  and  $\mu_y$  are the reduced effective masses in the AC and ZZ

directions, respectively), which is nearly layer-independent for 2-7L BP (*Nat. Commun.* 2014, **5**, 4475). Even so, our main conclusion is still not affected, that is “the continuum absorption near the band edge is almost a constant, independent of the thickness.”

To make the statements more properly, we made corrections in the revised main text and Supplementary Note 2. The main revisions are summarized as follows: (added sentences/words/equations are marked in red/yellow)

Page 9, Lines 184-199: “For *anisotropic* 2D massive semiconductors (see Supplementary Note 2 for details), based on the  $\frac{1}{k} \cdot \frac{\mathbf{u}}{p}$  perturbation theory34, it is

obtained that  $E_{cv}^{\mathbf{r}}(k) = E_g + \frac{\hbar^2}{2} \left( \frac{k_x^2}{\mu_x} + \frac{k_y^2}{\mu_y} \right)$ , and  $\frac{1}{\mu_x} = \frac{4 |P_{cv}^{\mathbf{u}}|^2}{m_0^2 E_g}$ , where  $\mu_{x(y)}$  is the *reduced effective mass in the AC (ZZ) direction*. With these, and also taking into account the degrees of spin and valley degeneracy of  $g_s$  and  $g_v$ , we thus have

$$A(\hbar\omega) = \frac{g_s g_v \pi \alpha}{2} \cdot \frac{E_g}{\hbar\omega} \cdot \sqrt{\frac{\mu_y}{\mu_x}} \cdot \Theta(\hbar\omega - E_g) \quad (2)$$

where  $\Theta(\hbar\omega - E_g)$  is the step function, describing the 2D joint DOS, and  $E_g$  is the bandgap. This indicates that *most of* the band parameter dependent terms are perfectly canceled out, leaving behind the frequency dependent term  $\omega$  and the band

*anisotropy term*  $\sqrt{\frac{\mu_y}{\mu_x}}$ . Our observations in few-layer BP can be well explained by Eq.

(2): (i) above the bandgap  $\hbar\omega > E_g$ , the absorption gradually decreases as the photon energy increases; (ii) near the bandgap  $\hbar\omega \approx E_g$ , with  $g_s = 2$ ,  $g_v = 1$  for BP20, Eq. (2)

gives a value of  $\pi\alpha \sqrt{\frac{\mu_y}{\mu_x}}$  (or equals to the optical conductivity  $\sigma_0 \sqrt{\frac{\mu_y}{\mu_x}}$ ), *slightly*

*deviating from the universal value  $\pi\alpha$* . The prefactor  $\sqrt{\frac{\mu_y}{\mu_x}}$  *originates from the band*

*anisotropy, which is nearly layer-independent for 2-7L BP17*. When  $\mu_x = \mu_y$ , it collapses to the isotropic case36; (iii) with the photon energy higher than the second subband transition ( $T_{22}$ ), additional contribution in absorption shows up with another step

height of  $\pi\alpha \sqrt{\frac{\mu_y}{\mu_x}}$ .”

- In line 121, the authors compare the absorption between traditional QWs and few-layer BP, arguing that the “hard confinement” in atomically thin BP distinguishes it from the QWs. What is the reason for the hard confinement in few-layer BP? Is it unique in few-layer BP?

Reply: For transitional QWs, taking  $\text{In}_x\text{Ga}_{1-x}\text{As}/\text{GaAs}$  QWs as an example, one  $\text{In}_x\text{Ga}_{1-x}\text{As}$  well layer is sandwiched between two GaAs barrier layers. Typically, the barrier is not high enough (finite), so the wavefunctions of carriers can leak into the barrier layer. While in this work, few-layer BP is supported on a PDMS or quartz substrate and with air (or vacuum) on its top. The carriers are strictly confined in the BP layer, with its work function as the barrier height. Although the work function is not infinite, but large enough ( $>4.56$  eV, Sci. Rep. 2014, 4, 6677) especially compared with shallow QWs. This is the so-called “hard confinement”. It’s not unique for few-layer BP and also applied to other exfoliated 2D materials.

To make the statement more properly, we modified sentences in the revised manuscript as follows: (added sentences/words are marked in red, deleted ones are marked by the strikethrough)

Page 6, Lines 130-132: “..., due to the exciton wavefunction leakage into the barriers in the ultrathin limit27, 28. *Such scenario becomes more evident in shallow QWs. The hard confinement in atomically thin BP definitely distinguishes it from traditional QWs.*”

- In Eq. (2) of the main text and Supplementary Note 2, the authors present the result of the universal absorption based on Supplementary Ref. 6. Is it valid for any massive 2D systems irrespective of their Dirac nature? It would be useful to discuss the validity of the result.

Reply: It’s an interesting question. In Supplementary Ref. 6 (PRB 2015, 91, 115407), the authors proposed a generalized formula of optical conductivity in isotropic 2D systems:  $\sigma = g_s g_v v \sigma_{\min}$ , where  $\sigma_{\min} = e^2/16\hbar = \sigma_0/4$  is a minimal universal conductivity defined by the authors, equal to 1/4 of the well-known universal conductivity of graphene ( $\sigma_0$ ),  $g_s$  and  $g_v$  denote the spin and valley degeneracy,  $v$  defines the curvature around the band gap ( $\epsilon_{c,v} \sim |k|^\nu$ ).

1) For semiconductor BP with parabolic band dispersion, the carriers behave as massive Schrodinger fermions. The interband transition occurs at  $\Gamma$  point of the Brillouin zone, with only one valley involved, thus  $g_v = 1$ , together with  $g_s = 2$ ,  $v = 2$ ,

and also the band anisotropy, we have  $\sigma = \sigma_0 \sqrt{\frac{\mu_y}{\mu_x}}$  near the bandgap ( $\hbar\omega \approx E_g$ );

2) For massive graphene, in which massive Dirac fermions are hosted. Since two valleys (K and K') are involved in interband transitions, thus  $g_v = 2$ , together with  $g_s = 2$ ,  $v = 2$ , it reads  $\sigma = 2\sigma_0$  near the bandgap ( $\hbar\omega \approx E_g$ ). The twice value comes from the valley degeneracy;

3) Theoretically, Baik et al. (Nano Lett. 2015, 15, 7788) proposed that in K atom doped few-layer BP, massless Dirac fermions (i.e., Dirac cones) emerge with linear band dispersions in all momentum directions. In this case,  $g_v = 2$ , since interband transitions can occur at the two Dirac points ( $k_D$  and  $-k_D$ ), while  $v = 1$  due to the linear energy dispersion, together with  $g_s = 2$ , and also the band anisotropy, we have

$$\sigma = \sigma_0 \sqrt{\frac{\mu_y}{\mu_x}}.$$

In the revised Supplementary Information, we expanded Supplementary Note 2 as follows:

*“Besides the common case for typical semiconductors discussed above, Stauber et al. proposed a generalized formula for optical conductivity in isotropic 2D systems7:  $\sigma = g_s \cdot g_v \cdot v \cdot \sigma_{min}$ , where  $\sigma_{min} = \sigma_0/4$  is a minimal conductivity defined by the authors, equal to 1/4 of the universal optical conductivity of graphene,  $v$  is related to the band curvature ( $\epsilon_{c,v} \sim |k|^v$ ,  $v = 2$  for parabolic dispersion, and  $v = 1$  for linear dispersion). Now, we consider three cases to check its validity, as summarized in Supplementary Table 1. Note that for massive Dirac fermions, the value is twice of  $\sigma_0$ , due to the valley degeneracy.”*

Supplementary Table 1. Optical conductivity in different 2D cases

| Type of carriers             | Examples     | $g_s$ | $g_v$ | $v$ | Band anisotropy              | $\sigma$                                       |
|------------------------------|--------------|-------|-------|-----|------------------------------|------------------------------------------------|
| Massive Schrodinger fermions | Few-layer BP | 2     | 1     | 2   | $\sqrt{\frac{\mu_y}{\mu_x}}$ | $\sigma = \sigma_0 \sqrt{\frac{\mu_y}{\mu_x}}$ |

|                         |                                   |   |   |   |                              |                                                |
|-------------------------|-----------------------------------|---|---|---|------------------------------|------------------------------------------------|
|                         |                                   |   |   |   |                              | $(\hbar\omega \approx E_g)$                    |
| Massive Dirac fermions  | Massive graphene                  | 2 | 2 | 2 | 1                            | $2\sigma_0 (\hbar\omega \approx E_g)$          |
| Massless Dirac fermions | K-doped few-layer BP 8 | 2 | 2 | 1 | $\sqrt{\frac{\mu_y}{\mu_x}}$ | $\sigma = \sigma_0 \sqrt{\frac{\mu_y}{\mu_x}}$ |

[8] Baik, S. S., Kim, K. S., Yi, Y. & Choi, H. J. Emergence of two-dimensional massless Dirac fermions, chiral pseudospins, and Berry's phase in potassium doped few-layer black phosphorus. *Nano Lett.* **15**, 7788 (2015).

- If possible, it could be interesting to apply a perpendicular electric field which could tune the energy gap even to a negative value inducing a phase transition, as demonstrated in the above mentioned Baik et al, *Nano Lett.* **15**, 7788 (2015) and Fig. 2 there. Or even in the insulator phase with a finite electric field, there appear additional interband transitions which are not allowed in the absence of an electric field, as demonstrated by Lin et al of Ref. 20. It could be interesting to study how the additional peaks and corresponding exciton/continuum absorptions evolve with the number of layers in the presence of a perpendicular electric field.

Reply: It's a very good suggestion and the gated experiments will be definitely interesting, with a lot of new discoveries expected. Nevertheless, it's very challenging to prepare gated devices based on few-layer BP, since BP is very sensitive and degrades too fast during the standard nano-fabrication process. If samples are not of high quality as those in this study, it's very hard to quantitatively determine the optical conductivity. Additionally, for IR transmittance measurements, transparent substrates are required, such as quartz, PDMS, BaF2 et al. These substrates are insulating, significantly increasing the difficulty in nano-fabrication. Therefore, unfortunately, we are unable to conduct gated experiments for few-layer BP at current stage. The reviewer's suggestion is certainly very helpful for us in future studies and we plan to try photocurrent spectroscopy based on FTIR as reported by Ju et al. for gated bilayer graphene (*Science* 2017, 358, 907). This configuration can work in the reflectance mode, thus opaque and doped Si/SiO2 substrate is allowed. This will make nano-fabrication more feasible.

---

### Reviewer #3 (Remarks to the Author):

The manuscript reported an increasing exciton absorption and a constant continuum absorption in few-layer BP as the layer number decreased. The latter finding has been extensively reported for various 2D structures, including 2D layered materials and traditional quantum wells. The former finding is remarkably different from other 2D layered materials due to quantum confinement and discrete subbands, which is of great interest to researchers in low-dimensional physics. The following concerns need to be addressed before consideration for publication.

Reply: We thank the reviewer for his/her careful review of our manuscript.

1. In Fig. 2, the width of the exciton peaks increased as the layer number decreased, which seems to contribute to the most increasing of the integrated absorption. Could the authors explain this?

Reply: It's true that thinner BP has larger exciton width, but we do not think that it's responsible for the increased exciton absorption. Let's first briefly discuss the exciton linewidth, which is closely related to the exciton lifetime. We observed an interesting phenomenon that exciton width increases with decreasing thickness. An example is shown in Fig. R11(b), this sample contains adjacent 2L and 5L flakes on the same PDMS substrate. IR measurements were performed for these two flakes simultaneously, the extinction spectrum clearly shows that the 2L has a larger  $E_{11}$  width than 5L. In Fig. R11(a), the exciton width is plotted as a function of  $E_{11}$  or  $E_{22}$  transition energy. It shows that the exciton width linearly increases with transition energy both for  $E_{11}$  and  $E_{22}$ , similar trend was also observed in 1D carbon nanotubes (PNAS 2014, 111, 7564; Nano Lett. 2008, 8, 87). Though the asymmetric broadening by exciton localization inevitably contributes to the exciton linewidth as well, the increased width is mainly due to increased phase space for electron-electron and electron-phonon interactions at higher energy (PNAS 2014, 111, 7564). Thinner BP exhibits higher transition energy, thus has larger width.

Next, we'd like to clarify that the increased width doesn't contribute to the increased absorption. A recent study by Horng et al. (arXiv: 1908.00884) showed that for typical semiconductors, in the incoherent region ( $\Gamma_{\text{sca}} \gg \Gamma_{\text{rad}}$ ) of light-matter

interactions, the integrated exciton absorption only depends on the radiative decays  $\Gamma_{\text{rad}}$ , rather than the scattering decays  $\Gamma_{\text{sca}}$ . As seen in Fig. R12, taking monolayer MoSe2 as an example, in the incoherent region, with increasing  $\Gamma_{\text{sca}}$ , the exciton width increases, but the integrated absorption (i.e., area of the exciton peak) maintains an approximately constant. This is also the case for few-layer BP, as shown in Fig. 4 of our manuscript, with increasing temperature, the exciton width increases, but the absorption area barely changes. This manifests that increased width in thinner BP is not responsible for the increased exciton absorption, instead the enhanced excitonic effects is the physical origin.

Fig. R11: Exciton linewidth as a function of transition energy in 2-7L BP.

Fig. R12: Absorption mechanism in an excitonic system (adapted from arXiv: 1908.00884).

2. In Fig. 3a, the authors added the substrate screening to decrease the slope. But the fitting is still not good enough for the more slowly descending data points.

Reply: In Fig. 3a of our manuscript, we can see that the modified model (red line) is much better than the  $1/N$  fitting (black line), but as the reviewer said, the agreement with experimental data is still not so excellent. The deviation is mainly caused by the

experimental uncertainty in determining the optical conductivity, for the following reasons: 1) BP samples are susceptible to the environment, especially for thinner ones. For example, excitonic effects will be reduced due to the dielectric screening by the charged impurities or defects from the substrate. For each exfoliated and measured sample, the PDMS substrate condition is not identical. 2) For IR measurements, large area samples are required, typically over  $20 \times 20 \mu\text{m}^2$ . In such large spatial range, the “inhomogeneity” is easily introduced, such as wrinkles in exfoliated samples, which will modify the optical absorption. Therefore, it’s very difficult to guarantee that different samples on different substrates are of the same quality. In view of this, we prepared a large amount of few-layer BP samples, only those with large area and optically flat surface are selected for IR measurements to reduce experimental errors. Even as careful as this, there is still inevitable experimental uncertainty, evidenced by the error bar (sample to sample variations) in Fig. 3a. Nevertheless, we believe that the overall trend is unambiguous. Our conclusion is robust that exciton absorption increases with decreasing thickness in few-layer BP, and that the substrate plays a role in reducing the exciton absorption. In the revised manuscript, to make the statement more clearly, we added the following comment:

Page 6, Lines 120-124: *“The basic behavior is well captured by the modified model, though the agreement with experimental data is still not so excellent. The deviation is mainly caused by the experimental uncertainty in determining the optical conductivity, especially for thinner BP samples, which are expected to be more susceptible to the environment.”*

3. For the vertical arrows, how did the authors choose the position of the arrows? The reviewer suggests to fit the continuum absorption and get the optical conductivity at the band edge, which would make Fig. 3b more convinced.

Reply: It’s indeed a good strategy to fit the exciton and continuum absorption together, then to extract the bandgap as a fitting parameter. Whereas, in practice, the fitting to continuum will be much complicated. Since a lot of uncertain factors will be involved, such as homogeneous broadening due to scattering at nonzero temperature, inhomogeneous broadening from the environment, and non-resonant

background caused by the substrate, et al. Alternatively, we determined the bandgap according to the formula  $E_g = E_{11} + E_b$ , where  $E_{11}$  is the 1s exciton peak energy and  $E_b$  is the exciton binding energy. Based on our previous study (Sci. Adv. 2018, 4, eaap9977),  $E_b$  is available for few-layer BP on PDMS substrate, expressed as

$$E_b = \frac{3}{4\pi(\chi_0 + N\chi_1)} \text{ in atomic units, with } \chi_0 = 6.5 \text{ \AA} \text{ and } \chi_1 = 4.5 \text{ \AA}, N \text{ is layer number.}$$

The values for 2-7L BP are summarized in Table 2. The “bandgap” related to  $E_{22}$  transition is determined similarly. Thus, the vertical arrows in Fig. 2 of our manuscript are showed to indicate the bandgap position. Since we determined the band edge conductivity in the same manner for different thickness, the comparison between them is fair and reliable. Therefore, our evidence is solid to support our conclusion that the continuum absorption is layer-independent.

Table 2: Exciton binding energy and bandgap of 2-7L BP on PDMS substrate

| N | $E_{11}$ (eV) | $E_b$ (eV) | $E_g$ (eV) |
|---|---------------|------------|------------|
| 2 | 1.11          | 0.22       | 1.33       |
| 3 | 0.85          | 0.17       | 1.02       |
| 4 | 0.70          | 0.14       | 0.84       |
| 5 | 0.61          | 0.12       | 0.73       |
| 6 | 0.54          | 0.10       | 0.65       |
| 7 | 0.50          | 0.09       | 0.59       |

4. For the title. “Less is more” is only correct for the increasing exciton absorption, not for the constant continuum absorption. The present title may produce misleading.

Reply: We fully agree with the reviewer on this issue. However, we want to emphasize the most important finding of our paper in the title, which is the exciton strength, so the current title was chosen.

Reviewers' comments:

Reviewer #1 (Remarks to the Author):

I'm fully satisfied with the authors' responses and performed revisions. Thus, I suggest accepting the revised paper.

Reviewer #2 (Remarks to the Author):

Overall, in the revised manuscript, the authors improved the manuscript following the comments and suggestions raised in the previous version of the manuscript.

One remaining question, however, is on the unique optical property of few-layer black phosphorus (BP). It seems the one of the main points of the current work is the observation of the enhanced exciton absorption with decreasing the number of layers due to the hard confinement in few-layer BP. In this point of view, only essential difference in few-layer BP compared to quantum wells is the hard confinement which does not lead to a leakage of wavefunctions, while in quantum wells there occurs the leakage in the thin film limit due to the finite barrier height. However, as the authors mentioned in the response, the hard confinement is "not unique for few-layer BP and also applied to other exfoliated 2D materials". In the manuscript, the authors compared only the few-layer BP, quantum wells and TMDCs, but there are various 2D materials which can be exfoliated.

- How are the hard confinement in other 2D materials?
- Does few-layer BP show the hardest confinement among 2D materials? If not, why is it special?
- Is it possible that other 2D materials having hard confinement show the enhanced exciton absorption similarly as in few-layer BP?
- In the application point of view, what is the implication of the enhanced oscillator strength with the decreasing thickness?
- Is the enhanced oscillator strength with decreasing the number of layers first observed by the authors in few-layer BP?

Still, it is not clear even in the revised manuscript why few-layer BP is unique in optical properties compared to other 2D materials. I want the authors to clarify this point in the manuscript.

Reviewer #3 (Remarks to the Author):

All major concerns from the reviewer have been addressed, the manuscript could be published in Nature Communications.

## Response to the reviewers' comments

~~~~~

Reviewer #1 (Remarks to the Author):

I'm fully satisfied with the authors' responses and performed revisions. Thus, I suggest accepting the revised paper.

Reply: We thank the reviewer again for his/her careful review of our manuscript and recommendation of publication in Nature Communications.

~~~~~

### Reviewer #3 (Remarks to the Author):

All major concerns from the reviewer have been addressed, the manuscript could be published in Nature Communications.

Reply: We thank the reviewer again for his/her careful review of our manuscript and recommendation of publication in Nature Communications.

~~~~~

Reviewer #2 (Remarks to the Author):

Overall, in the revised manuscript, the authors improved the manuscript following the comments and suggestions raised in the previous version of the manuscript.

One remaining question, however, is on the unique optical property of few-layer black phosphorus (BP). It seems the one of the main points of the current work is the observation of the enhanced exciton absorption with decreasing the number of layers due to the hard confinement in few-layer BP. In this point of view, only essential difference in few-layer BP compared to quantum wells is the hard confinement which does not lead to a leakage of wavefunctions, while in quantum

wells there occurs the leakage in the thin film limit due to the finite barrier height. However, as the authors mentioned in the response, the hard confinement is “not unique for few-layer BP and also applied to other exfoliated 2D materials”. In the manuscript, the authors compared only the few-layer BP, quantum wells and TMDCs, but there are various 2D materials which can be exfoliated.

- How are the hard confinement in other 2D materials?

Reply: We believe that for other exfoliated 2D materials supported by dielectric substrates or in free-standing case, the hard confinement still holds as that in few-layer BP, since carriers are strictly confined in the layer plane, and almost impossible to tunnel into its surroundings.

- Does few-layer BP show the hardest confinement among 2D materials? If not, why is it special?

Reply: As mentioned above, the hard confinement is not unique for few-layer BP, and also applied to other exfoliated 2D materials. However, it's unique in the sense that it has clean manifestation of this hard confinement effect in the exciton oscillator strength, while such effect in other 2D semiconductors is obscured due to various complications. The possible candidates which might show similar “less is more” effect should be direct gap 2D semiconductors regardless of the thickness, with well separated subbands. The closest well studied example we can think of is InSe. However, as shown in Fig. R1, the enhanced exciton oscillator strength in thin samples is obscured by strong backgrounds. Admittedly, since there are so many potentially exfoliated 2D materials, there will be other 2D semiconductors with similar layer-dependent excitonic properties. Nevertheless, BP is the first such material to manifest the effect and the significance of our work is not compromised.

Fig. R1: The differential reflectance spectra of 1- to 8-layer InSe, the gray triangles denote the absorption peak of exciton B (adapted from PRB 2019, 99, 195414).

- Is it possible that other 2D materials having hard confinement show the enhanced exciton absorption similarly as in few-layer BP?

Reply: The exciton absorption is mainly determined by the joint DOS or band structure. Taking MoS₂ as an example, which is believed to have hard confinement as well, but actually the exciton absorption is reduced in thinner samples, since less nearly degenerate subbands are involved in the optical transition, the DOS is consequently decreased. If a 2D material has the same DOS for different layer number, more straightforwardly, the continuum absorption is the same, we believe that it will show enhanced absorption similarly as in few-layer BP. A potential candidate is black arsenic, which is yet to be explored.

- In the application point of view, what is the implication of the enhanced oscillator strength with the decreasing thickness?

Reply: Since thinner samples absorb and emit more light, we can expect an enhanced efficiency for photodetectors and emitters made of thinner materials than thicker ones. For example, two artificially stacked "1L" BP (without interaction) is expected to exhibit markedly enhanced efficiency than one natural "2L" BP. This opens an avenue for more efficient optoelectronic devices.

- Is the enhanced oscillator strength with decreasing the number of layers first observed by the authors in few-layer BP?

Reply: To the best of our knowledge, it is the first experimental observation of enhanced oscillator strength with decreasing layer number in few-layer BP.

Still, it is not clear even in the revised manuscript why few-layer BP is unique in optical properties compared to other 2D materials. I want the authors to clarify this point in the manuscript.

Reply: We thank the reviewer for raising this point. To be more clarified, we made modifications in the revised manuscript as follows:

Page 2-3, Line 44-49: *“Moreover, the intrinsic in-plane band anisotropy definitely distinguishes few-layer BP from other widely studied 2D materials, such as graphene, TMDCs and InSe. Along with the moderate bandgap and large tunability, few-layer BP is unique and promising in polarized IR detectors and emitters. From this point of view, a quantitative determination and thorough understanding of the optical absorption in few-layer BP is in great demand.”*